# The burden and trends of child and maternal malnutrition across the regions in Ethiopia, 1990–2019: The Global Burden of Disease Study 2019

**Mesfin Agachew Woldekidan**[1]*, **Asrat Arja**[1], **Getaye Worku**[1], **Ally Walker**[2], **Nicholas J. Kassebaum**[2], **Alemnesh Hailemariam**[1], **Mohsen Naghavi**[2], **Simon Hay**[2], **Awoke Misganaw**[2]

1 National Data Management and Analytics Center, Ethiopian Public Health Institute, Addis Ababa, Ethiopia, 2 Institute for Health Metrics and Evaluation, University of Washington, Seattle, Washington, United States of America

* mesfinagachew@gmail.com

## Abstract

### Introduction

Child malnutrition is the main contributor to the disease burden in Ethiopia. The objective of this study was to determine the prevalence and trends of child malnutrition and maternal anemia in Ethiopia at the national and regional state levels between 1990 and 2019.

### Methods

We used all accessible data sources and analyzed prevalence, death, and years of life lost (YLL) due to child malnutrition and maternal anemia across nine regions and two chartered cities in Ethiopia, as part of the Global Burden of Diseases, Injuries, and Risk Factors Study (GBD) 2019. The burden and trends of child and maternal malnutrition and anemia at the national level, across the regions, and in cities were assessed. Point estimates with 95% uncertainty intervals (UI) are presented.

### Findings

Of the 190,173 total under-5 deaths in Ethiopia in 2019, 108,864 (95% UI: 83,544–141,718; 57·2%, 51·3–62·7) were attributed to malnutrition. The prevalence of stunting, underweight, and wasting was 37·0%, 27·0%, and 7·0%, respectively, in 2019. The YLL rate attributable to child malnutrition declined from 251,964 per 100,000 population (95% UI: 218,720–287,559) in 1990 to 57,615 (95% UI: 44,190–75,015) in 2019. The YLL rate of wasting, stunting, and underweight in Ethiopia was 18,566 per 100,000 population (95% UI: 12,950–26,123), 3,290 (95% UI: 1,443–5,856), and 5,240 (95% UI: 3,608–7,312) in 2019, respectively. Gambella showed the highest YLL rate reduction among regions, with a 98·2% change for stunting, 95·9% for wasting, and 97·9% for underweight between 1990 and 2019. The prevalence of anemia among under-5 children in Ethiopia was 62·0% (95% UI:

**Data Availability Statement:** The data underlying the results presented in the study are available

from the Global Burden of Diseases Study (GBD). The GBD provides a comprehensive set of health-related data that is publicly accessible. The full GBD results data, including the data used in this study, can be accessed in CSV format from the GBD Results Tool here (http://ghdx.healthdata.org/gbd-results-tool).

**Funding:** The Bill and Melinda Gates Foundation has provided funds for EPHI and IHME collaborative Ethiopia subnational burden of disease study initiative. The funders had no role in study design, data collection and analysis, decision to publish, or preparation of the manuscript.

**Competing interests:** The authors have declared that no competing interests exist.

59·1%–65·1%) in 2019. Somali has the highest child anemia prevalence, 84·4% (95% UI: 79·8%–88·8%), compared to others in 2019. The prevalence of anemia in women of reproductive age (15–49 years) in Ethiopia was 20·4% (95% UI: 19·0%–21·8%) in 2019.

## Interpretation

The prevalence of child malnutrition and maternal anemia in Ethiopia remains high compared to national, WHO, and UNICEF 2030 targets in all indicators of child malnutrition and anemia despite several interventions in the last three decades. The YLL rate due to child malnutrition was high, with regional variations. In conjunction with other sectors, especially agriculture, the National Nutrition Program and other nutrition initiatives must make greater efforts with short-term and long-term interventions to improve access and better nutrition.

## Introduction

Malnutrition refers to energy and/or nutrient deficits, excesses, or imbalances in a person's diet [1]. Child malnutrition is a neurotic state coming about from both undernutrition and overnutrition [2]. The insufficient intake of energy and other nutrients is called child undernutrition. Even though the term "child malnutrition" includes both undernutrition and overnutrition, we used it only for expressing child undernutrition in this paper. Maternal and child malnutrition affected more than 55% of the world's population in 2019 [3]. Globally, the rate of child malnutrition is consistently high; stunting is gradually decreasing, while wasting still impacts the lives of numerous young children [4]. High rates of malnutrition in young children, a condition that is strongly linked to poverty, persist worldwide [2]. Globally, 149 million children under the age of 5 are stunted, 49·5 million are wasted, and 40·1 million are overweight [5]. In 2017, only roughly one-quarter of the 16·6 million under 5 children in the world who had severe acute malnutrition received treatment, highlighting the critical need to address this unacceptable burden [6].

Most women living in developing countries experience various biological and social stresses that foster the danger of malnutrition for mothers and their children [7]. Women who suffer from malnutrition at the onset of pregnancy are unlikely to enhance their nutritional condition throughout the pregnancy [8]. Maternal undernutrition can be categorized as having a body mass index of less than 18·5, and maternal anemia as women having hemoglobin (Hb) levels less than 12 g/dL [9].

In African countries, child malnutrition is still a significant influence on child health and survival [10]. Child underweight continues to be a problem in the poorest countries, with rates up to 10 times greater than in the richest. Obesity and overweight are prevalent in the richest countries, with rates up to five times greater than in the poorest [6]. Africa and Asia have the highest share of malnutrition in the globe [11].

Like child malnutrition, maternal malnutrition in Ethiopia is also much higher than in many African countries [12]. By eliminating undernutrition, Ethiopia would prevent losses of 8% to 11% per year from the gross national product [13]. Maternal anemia, stunting, wasting and underweight are a persistent national challenge with serious consequences for survival and the occurrence of acute and chronic diseases, negatively affecting healthy development and financial production at both individual and societal levels [14]. The rise in diet-related noncommunicable diseases is increasingly putting pressure on health systems in Ethiopia [15].

Worldwide, about one in three women of reproductive age (15–49) is estimated to suffer from anemia, presenting with a hemoglobin level below 11 g/dL [16]. Adult labor capacity was linked to maternal anemia, and pregnancy-related anemia which leads to high risk of maternal death and low birthweight [17].

Although some small-scale studies in Ethiopia have reported the trends of some of the indicators used for monitoring progress and the burden of malnutrition [18], a comprehensive study has not been done showing both the national and subnational burden beside the trends over time, mainly due to data gaps. For nations like Ethiopia that struggle to access complete and comparable data for measuring indicators and tracking progress over time, the Global Burden of Disease Study (GBD) 2019 has presented a unique opportunity. With GBD 2019, Ethiopia has estimates for the indicators used for measuring malnutrition among women and children for the first time at the level of regional states and chartered cities.

In Ethiopia, various approaches have been applied to reverse the increasing prevalence of child and maternal malnutrition. Ethiopia was one of the first nations called early risers to participate in the Climb up Nutrition Movement, and the 2008 *Lancet* series provided the science foundation for the movement, with a focus on the first one thousand days [19]. Before the first National Nutrition Program (NNP) was created, the majority of nutrition interventions in Ethiopia focused mostly on emergency feeding and micronutrient supplementation and very little on prevention [20]. In addition, Ethiopia made the Seqota declaration on July 15, 2015, to end child malnutrition by 2030, reaffirming its commitment to nutrition as a foundation for economic development [21]. Thus, the objective of this analysis is to evaluate the disease burden caused by malnutrition, besides trends over time, at the national and regional levels in Ethiopia.

## Methods

### Study setting

Ethiopia is the second-most populous country in Africa with an estimated population of 112 million in 2019 [22]. Ethiopia is home to about 13 million children under 5 years of age–approximately 16% of the total population [23]. Ethiopia is composed of 11 regions (Afar; Amhara; Benishangul-Gumuz; Gambella; Harari; Oromia; Somali; Sidama; Southern Nations, Nationalities and People [SNNP], Tigray; and South Western Ethiopia) and two chartered cities (Addis Ababa and Dire Dawa) under a federal system of administration [24]. Oromia, Amhara, and SNNP are highly populous regions. Chartered cities means, collectively, Addis Ababa City Government and Dire Dawa City Administration, as defined in their federal charter proclamations, or any successor or successors thereto; and "chartered city" means any of the above chartered cities [25]. The current administrative system consists of the federal government, regions/chartered cities, zones, woredas or "districts," and kebeles (lowest administrative units). In this analysis, the Sidama region and Southwestern Ethiopia, which were established in 2020–2021, were included under Southern Nations and Nationalities Peoples region as the data were taken in 2019, prior to the establishment of Sidama region of Southwestern regions. Ethiopia's health care is a three-tiered system which includes primary, secondary, and tertiary levels of care with 21,154 functioning health facilities and 159,545 health workers in 2019 [26].

### Overall GBD methods and tools

The details of the GBD methodology have been reported elsewhere [27]. The details on data sources and methods in Ethiopian subnational GBD analysis were reported in the health progress study in Ethiopia [28]. We summarized methods and tools relevant to this analysis. The

GBD used a modeling platform called the Cause of Death Ensemble model (CODEm) to estimate child and maternal mortality by age, sex, geography, and year. Child and maternal morbidity, including incidence, was modeled using a meta-regression platform known as disease modeling meta-regression (DisMod-MR), a Bayesian, hierarchical, mixed-effects meta-regression. Years of life lost (YLLs) were computed by multiplying cause-specific deaths by the life expectancy at the age of death. Then, years lived with disability (YLD) and YLLs were added to calculate disability-adjusted life years (DALYs) for each regional state by sex, year, and age group. Population risk evaluations over time and among risks were estimated using the comparative risk assessment (CRA) approach created for the GBD study. GBD diseases and injuries were organized into a leveled cause hierarchy from the three biggest causes of death and disability at Level 1 to the foremost causes at Level 4. There were 174 Level 3 causes among the three Level 1 causes (communicable, maternal, neonatal, and nutritional disease; non-communicable diseases; and injuries). The GBD risk factors were also organized into a leveled hierarchy; Level 1 risk factors are behavioral, environmental and occupational, and metabolic; Level 2 risk factors include 20 clusters of risks; Level 3 includes 52 clusters of risks; and Level 4 includes 69 specific risk factors. GBD 2019 has estimated the burden of child and maternal malnutrition for both national and regional states of Ethiopia.

### Estimation of exposure to malnutrition and operational definitions

Malnutrition exposure and disease burden were estimated using the GBD comparative risk assessment methodology to report the prevalence of the eight malnutrition indicators included in Ethiopia, and compared with the World Health Organization (WHO) and United Nations Children's Fund (UNICEF) target 2025 [9]. For the malnutrition indicators, the following definitions were used: low birthweight is taken as less than 2,500 g; stunting in children under 5 years taken as height for age and length for age, wasting taken as weight for height, and underweight as weight for age below two standard deviations (SDs) of the median in the WHO 2006 standard curve; anemia in children younger than 5 years as hemoglobin less than 110 g/L; and anemia in women 15–49 years of age considered as hemoglobin less than 110 g/L if pregnant and 120 g/L if not pregnant [29].

### Estimation of prevalence and YLLs attributable to malnutrition

Estimation of attributable disease burden included determining the relative risk of disease outcomes for risk exposure–disease outcome pairs with sufficient evidence of a causal relationship in randomized controlled trials and prospective cohort studies, as assessed using a grading system similar to that used by the World Cancer Research Fund [30]. Estimates of death and YLLs attributed to each malnutrition risk factor were calculated using population attributable fractions by location, age, and year. The general equation and formula can be found elsewhere [31].

All malnutrition indicator estimates generated in GBD 2019 were accompanied by 95% uncertainty intervals (UIs). The uncertainty ranges stated around YLDs include uncertainties in both prevalence and disability weight. The UI was created using 1,000 model runs for each quantity of interest, which are sufficient for the GBD models [30]. The 95% UI is described as the 25th and 975th values of the distribution.

### Presentation and interpretation of results

We presented findings on the burden of maternal anemia and child malnutrition and trends over time for the nation and the nine regional states in Ethiopia (Afar, Amhara, Benishangul-Gumuz, Gambella, Harari, Oromia, Somali, SNNP, and Tigray), and two chartered cities,

Addis Ababa and Dire Dawa [23]. We used the GBD 2019 categorization of diseases and injuries to show and interpret the national and regional prevalence and YLL rates of common child malnutrition indicators in Ethiopia from 1990 to 2019. We summarized the trends of child malnutrition indicators from 1990 to 2019 in terms of YLLs, and prevalence rates whenever available. We presented percentage change from 1990 to 2019 and Socio-demographic Index (SDI). The SDI is the geometric mean of 0 to 1 indices of the total fertility rate younger than 25 years, the mean education for those aged 15 years and older, and lag-distributed income per person [28]. We assessed the rates of deaths and YLL rates attributable to child malnutrition among children under 5 years of age and the prevalence of maternal anemia in the reproductive age group in all regional states in 2019 and compared them with other risk factor categories. Anemia prevalence and percentage change were reported in children under 5 years of age and women of reproductive age (15–49 years) in Ethiopia and regional states from 1990 to 2019.

### Role of the funding source

The Bill and Melinda Gates Foundation financed this subnational analysis. The study's funders were not involved in the study's design, data collection, data analysis, data interpretation, or report writing. The corresponding authors had full access to all the study's data and were ultimately in charge of deciding whether or not to publish it.

### Ethical approval and consent to participate

The University of Washington's Institute for Health Metrics and Evaluation (IHME) health data portal gave open access GBD 2019 secondary data for the manuscript.

## Results

### SDI and the burden of malnutrition in children

In 2019, the SDI for Ethiopia was 0.34. Addis Ababa had the highest SDI (0.64), followed by Dire Dawa, Harari, and Gambella. Somali (0.19) and Afar (0.26) had the lowest SDIs. The SDI increased in all regions and cities between 1990 and 2019. The SDI of Ethiopia rose steadily from 0.13 in 1990 to 0.34 in 2019 (Table 1). As stunting, wasting, and underweight decreased from 1990 to 2019, the SDI also decreased.

Of the 190,173 total under-5 deaths in Ethiopia in 2019, 108,864 (95% UI: 83,544–141,718; 57·2%, 51·3–62·7) were attributed to malnutrition. All-cause under-5 death rates attributed to malnutrition in Ethiopia decreased significantly, from 2,874 per 100,000 (95% UI: 2,494–3,281) in 1990 to 653 per 100,000 (95% UI: 501–850) in 2019. Child death rates due to malnutrition showed a decline of 77·3% between 1990 and 2019 (Fig 1).

As shown in Fig 1, SNNP and Benishangul-Gumuz had substantially higher under-5 death rates per 100,000 population in 2019. The change rate for Gambella showed a progression of 88·1% from 1990 to 2019, which was better progress compared to other regions. Next to Gambella, Addis Ababa also showed a decrement of 87·4% between 1990 and 2019. SNNP showed very low progression between 1990 and 2019 of only 57·0% (Fig 1).

In Ethiopia, the YLL rate attributable to child malnutrition was 57,615 (95% UI: 44,190–75,015) in 2019, which was lower compared to the sub-Saharan Africa average of 73,529 (95% UI: 59,587–92,393) and much higher than the global average of 37,010 (95% UI: 31,028–44,608) (Fig 2). Ethiopia had a significant decrement, 77·1%, between 1990 and 2019, and showed a low YLL rate starting in 2005 compared to the sub-Saharan Africa region. Compared to global estimates, Ethiopia and sub-Saharan Africa had higher rates of YLLs attributed to

**Table 1. Years of life lost (YLLs) rate of stunting, wasting, underweight and low birth weight in under-five children in all regions and chartered cities of Ethiopia, 1990–2019.**

| Regions | Stunting | | | Wasting | | | SDI | |
|---|---|---|---|---|---|---|---|---|
| | 1990 (95% UI) | 2019 (95% UI) | % of Change | 1990 (95% UI) | 2019 (95% UI) | % of Change | 1990 | 2019 |
| Addis Ababa | 8142.2 (1913.9–64981.8) | 173.7 (56.7–396.8) | -97.9% | 46492.2 (31118.7–62149.1) | 2234.1 (1196.4–3625.2) | -95.2 | 0.38 | 0.64 |
| Afar | 29692.3 (22408.2–90607.8) | 2603.8 (1054.8–4900.9) | -91.2% | 111722.4 (80592.8–147324.9) | 15168.6 (7883.3–22789.4) | -86.4 | 0.08 | 0.26 |
| Amhara | 46460.2 (3453.9–16352.4) | 3897.2 (1603.9–7370.6) | -91.6% | 138893.4 (101108.9–189857.1) | 17540.9 (9994.3–28006.7) | -87.4 | 0.10 | 0.31 |
| Benishangul-Gumuz | 48713.7 (14335.6–54027.6) | 4427.1 (1669.4–8895.9) | -90.9% | 166910.7 (114301.8–228688.5) | 26777.3 (16195.2–39424.5) | -84.0 | 0.10 | 0.32 |
| Dire Dawa | 28602.1 (13697.2–60191.8) | 1018.9 (266.9–2461.2) | -96.4% | 133032.2 (87154.3–181111.0) | 9820.4 (4223.2–17173.8) | -92.6 | 0.26 | 0.50 |
| Gambella | 35878.7 (14779.1–52322.7) | 635.0 (180.3–1385.8) | -98.2% | 157606.2 (106249.7–211880.5) | 6419.5 (2807.0–10669.9) | -95.9 | 0.16 | 0.41 |
| Harari | 32368.6 (8742.2–43352.0) | 1353.8 (425.0–3154.7) | -95.8% | 129160.6 (88964.3–173349.0) | 10204.5 (4717.4–17398.6) | -92.1 | 0.27 | 0.49 |
| Oromia | 38008.5 (24392.1–87172.9) | 3242.6 (1320.7–6007.9) | -91.5% | 149560.2 (114521.8–188908.4) | 19781.2 (13232.8–27246.4) | -86.8 | 0.11 | 0.32 |
| SNNP | 49652.1 (25147.7–79491.0) | 3638.4 (1241.8–6214.8) | -92.7% | 140003.3 (94881.0–195336.5) | 17896.4 (10440.9–26448.3) | -87.2 | 0.11 | 0.33 |
| Somali | 20834.1 (12372.5–54685.3) | 3131.0 (1501.3–6633.9) | -85.0% | 86998.8 (58578.3–123946.8) | 27691.8 (17970.7–38602.4) | -68.2 | 0.04 | 0.19 |
| Tigray | 30105.6 (15312.8–67815.6) | 1577.4 (634.8–3003.4) | -94.8% | 116746.0 (86923.5–155235.5) | 9020.5 (5081.7–13377.1) | -92.3 | 0.13 | 0.36 |
| Ethiopia | 40142.8 (20776.1–70013.5) | 3290.0 (1442.7–5856.0) | -91.8% | 135174.9 (102738.9–177672.4) | 18565.7 (12950.1–26123.4) | -86.3 | 0.13 | 0.34 |

UI = uncertainty Intervals, SNNPR = Southern Nations and Nationalities, SDI = Socio Demographic Index

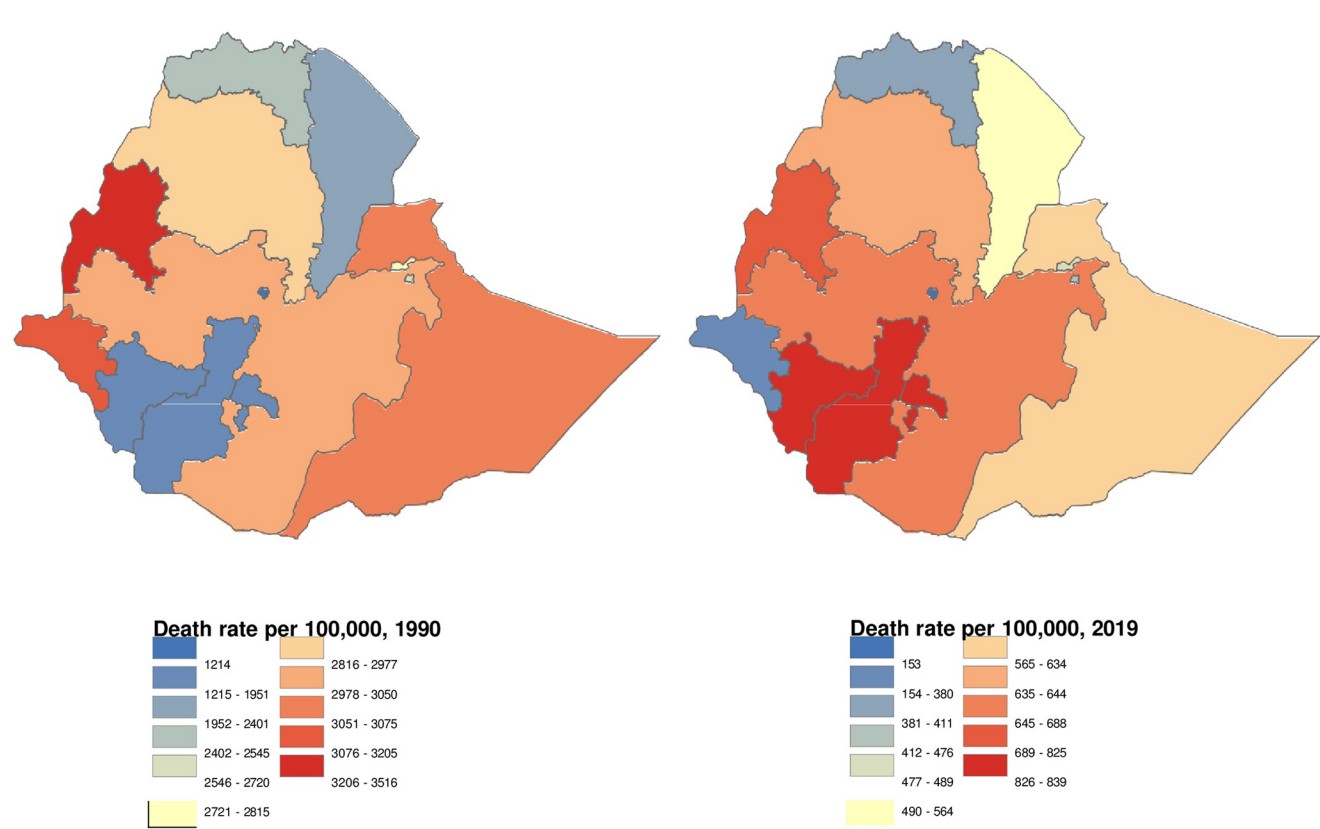

**Fig 1. Death rate per 100,000 population attributable to malnutrition for children under 5 years in Ethiopian regional states, 1990 and 2019 sourced from Natural Earth (http://www.naturalearthdata.com/about/terms-of-use/).**

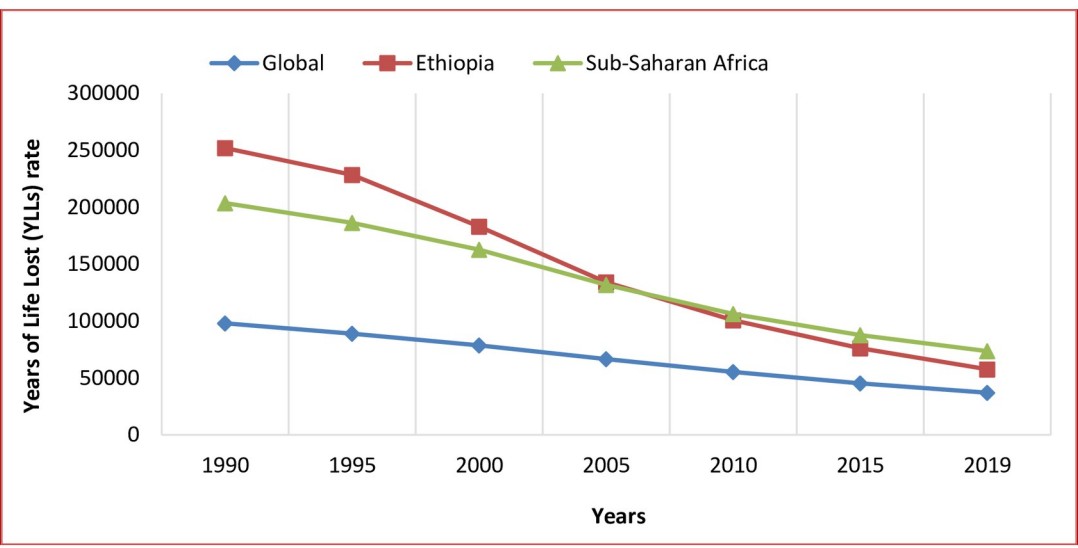

**Fig 2. The trend of years of life lost (YLLs) rate per 100,000 population attributable to malnutrition in Ethiopia, sub-Saharan Africa, and global, 1990–2019.** Note: Malnutrition here refers undernutrition.

child malnutrition from 1990 to 2019. The global YLL rate showed a consistent decline from 1990 to 2019 (Fig 2).

The YLL rate attributable to child malnutrition varied significantly across regions and cities in Ethiopia in 2019. Addis Ababa showed the lowest YLL rate attributable to malnutrition in children compared to other regions in 2019. In 2019, Somali and Benishangul-Gumuz had a YLL rate attributable to malnutrition in children under 5 years of age five times higher than Addis Ababa (Fig 3).

## Child stunting

The YLL rate attributable to stunting in children in Ethiopia was 3,290 per 100,000 population (95% UI: 1,443–5,856) in 2019. The YLL rate of child stunting has shown a substantial decrease between 1990 and 2019 in all the regional states. There are disparities in the YLL rate across regional states. Benishangul-Gumuz reported the highest YLL rate, 4,427 (95% UI: 1,669–8,896), whereas Addis Ababa, the capital of Ethiopia, had the lowest YLL rate for stunting, 174 (95% UI: 56·7–397). Gambella showed a 98·2% reduction of YLLs in 2019 compared to 1990, the highest reduction in the country, while the Somali region had an 85·0% reduction over the past 29 years (Table 1).

## Child wasting

The YLL rate of wasting in under-5 children in Ethiopia was 18,566 per 100,000 population (95% UI: 12,950–26,123) in 2019. There were disparities in the YLL rate of wasting across the regional states. Somali showed the highest YLL rate in 2019, 27,692 (95% UI: 17,971–38,602), whereas Addis Ababa recorded the lowest rate, 2,234 (95% UI: 1,196–3,625). There was a marked decrease between 1990 and 2019 in all regional states of Ethiopia except Somali and Benishangul-Gumuz, which had a lower reduction over the past 29 years compared to others. Gambella showed a 95·9% reduction in the YLL rate of wasting between 1990 and 2019, the

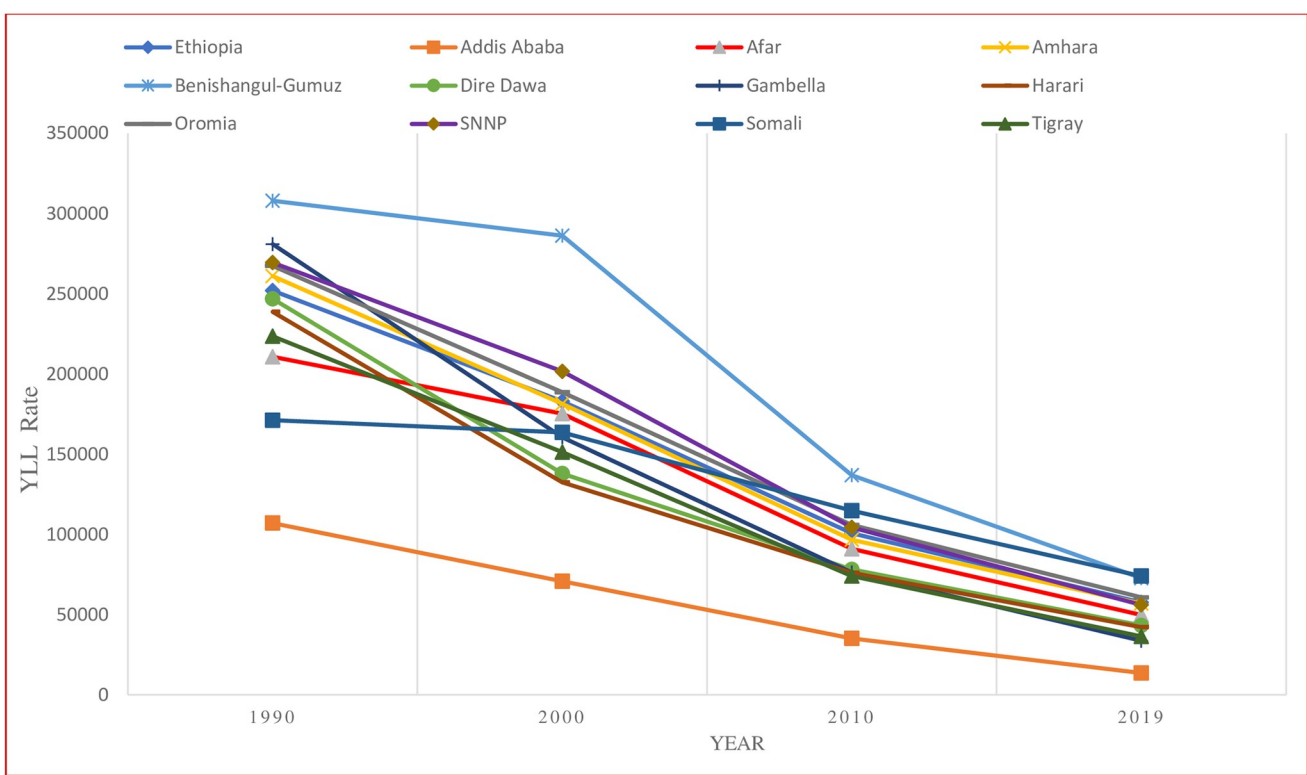

**Fig 3. Trend of years of life lost (YLLs) rate per 100,000 population attributable to malnutrition in children younger than 5 years in the regions of Ethiopia, 1990–2019.** Note: Malnutrition here refers undernutrition.

highest in the country, followed by Addis Ababa, while Somali recorded only a 68·2% reduction over the past 29 years (Table 1).

## Child underweight

The YLL rate of underweight in Ethiopia was 5,240 per 100,000 population (95% UI: 3,608–7,312) in 2019. Addis Ababa had the lowest YLL rate compared to others, 438 (95% UI: 222–761), and Somali had the highest, 9,763 (95% UI: 5,916–14,395) in 2019. A considerable decrease occurred between 1990 and 2019 in all regional states of Ethiopia except Somali and Benishangul-Gumuz, which had a lower reduction over the past 29 years compared to others (Table 2).

## Low birthweight

The YLL rate of low birthweight in Ethiopia was 34,070 per 100,000 population (95% UI: 26,84–43,407) in 2019. Addis Ababa had the lowest YLL rate compared to others, 10,148 (95% UI: 6,966–14 313), and Somali had the highest, 41,224 (95% UI: 32,587–52,592) in 2019. A decrease in YLLs was seen between 1990 and 2019 in all regional states of Ethiopia but was low compared to stunting, wasting, and underweight. Somali and Oromia regions had a lower reduction over the past 29 years compared to others (Table 2).

## Prevalence of stunting, wasting, and underweight

The prevalence of child stunting in Ethiopia was 37·0% in 2019, showing a 36·0% decrement from the 2000 baseline (Fig 4). The prevalence of child wasting in Ethiopia was 7·0% in 2019,

**Table 2. Years of life lost (YLLs) rate of underweight and low birth weight in under-five children and percentage of change in all regions and chartered cities of Ethiopia, 1990–2019.**

| | Underweight | | | Low Birth Weight | | | SDI[7] | |
|---|---|---|---|---|---|---|---|---|
| | 1990 (95% UI) | 2019 (95% UI) | % Change 1990 to 2019 | 1990 (95% UI) | 2019 (95% UI) | % Change 1990 to 2019 | 1990 | 2019 |
| **Addis Ababa** | 15612.2 (9992.1–22607.7) | 438.3 (222.1–761.2) | -97.2% | 47870.2 (37913.1–59278.4) | 10147.9 (6965.6–14312.7) | -85.4% | 0.38 | 0.64 |
| **Afar** | 58566.6 (39394.6–79453.8) | 5123.6 (2566.9–7848.7) | -91.3% | 71735.5 (61150.2–83072.6) | 30971.8 (23733.3–40899.2) | -66.9% | 0.08 | 0.26 |
| **Amhara** | 72597.4 (49966.2–102872.1) | 4486.8 (2632.4–7067.1) | -93.8% | 81963.6 (70274.7–94248.8) | 33817.2 (26106.6–43728.2) | -68.1% | 0.10 | 0.31 |
| **Benishangul Gumuz** | 87392.4 (57308.3–125692.9) | 9454.5 (5607.5–14094.2) | -89.2% | 87318.3 (73490.2–103318.0) | 38516.3 (29232.2–50338.8) | -66.5% | 0.10 | 0.32 |
| **Dire Dawa** | 58406.3 (36617.9–85170.0) | 2481.0 (1040.6–4460.7) | -95.8% | 81883.3 (67501.6–96305.8) | 29354.0 (22112.9–38636.6) | -73.0% | 0.26 | 0.50 |
| **Gambella** | 62361.8 (40712.6–89635.8) | 1314.3 (571.1–2362.5) | -97.9% | 88395.8 (75160.9–102887.0) | 24596.2 (18154.8–32354.7) | -79.5% | 0.16 | 0.41 |
| **Harari** | 45031.8 (28826.2–65007.9) | 2321.4 (1035.0–4113.7) | -94.8% | 74930.8 (61722.9–89494.9) | 26882.3 (20181.1–35351.5) | -73.1% | 0.27 | 0.49 |
| **Oromia** | 75719.2 (53163.4–103867.5) | 5345.6 (3481.6–7590.5) | -92.9% | 82925.0 (72985.2–93869.1) | 36294.1 (28713.3–46180.6) | -65.4% | 0.11 | 0.32 |
| **SNNP** | 81631.1 (53999.8–116497.8) | 5199.0 (3015.2–7785.0) | -93.6% | 79793.0 (68577.7–91234.9) | 32250.4 (24984.4–42313.4) | -68.7% | 0.11 | 0.33 |
| **Somali** | 44924.9 (27394.6–67555.9) | 9762.6 (5915.5–14395.2) | -78.3% | 58891.8 (50659.1–66988.2) | 41224.3 (32587.4–52592.4) | -44.7% | 0.04 | 0.19 |
| **Tigray** | 64665.3 (44225.6–90894.5) | 2186.7 (1216.1–3404.0) | -96.6% | 76335.8 (65585.0–87394.3) | 24135.5 (18670.2–31543.3) | -75.5% | 0.13 | 0.36 |
| **Ethiopia** | 71364.1 (51315.2–96951.5) | 5240.1 (3607.8–7312.4) | -92.7% | 78886.4 (69279.3–89285.7) | 34069.9 (26839.7–43406.9) | -66.0% | 0.13 | 0.34 |

UI = uncertainty Intervals, SNNPR = Southern Nations and Nationalities, SDI = Socio Demographic Index

showing a 41·6% decrement from the 2000 baseline. The highest reduction, 30·0%, was recorded between 2016 and 2019. The prevalence of wasting in Ethiopia was 7·0% in 2019, with no decrease between 2000 and 2005, or between 2011 and 2016. Instead, the highest decline, 30·0%, occurred between 2016 and 2019. Minimal decrement showed in the prevalence of stunting between 2016 and 2019, only 2·6%. The highest reduction, 19·5%, was recorded between 2000 and 2005 (Fig 4).

## Child anemia

The prevalence of anemia in children under 5 in Ethiopia was 62·0% (95% UI: 59·1%–65·1%) in 2019 and did not show a marked decrease between 1990 and 2019 (Table 3). In 2019, Somali had an anemia prevalence of 84·4% (95% UI: 79·8%–88·8%), the highest in the country. Afar and Dire Dawa had the second and the third highest, with a prevalence of 78·8% (95% UI: 73·1%–84·4%) and 74·1% (95% UI: 68·9%–79·3%), respectively (Table 3).

## Anemia in women

The prevalence of anemia in women of reproductive age (15–49 years) in Ethiopia was 20·4% (95% UI: 19·0%–21·8%) in 2019 and did not show a substantial decrease over the past 29 years (Table 3). Somali had an anemia prevalence of 45·0% (95% UI: 40·8%–48·1%), the highest in the country. Afar and Dire Dawa were the second and the third with prevalence of 36·1% (95% UI: 32·9%–39·1%) and 28.5% (95% UI: 25·1%–32·0%), respectively. Addis Ababa, Amhara, and Benishangul-Gumuz showed a 34·4%, 33·9%, and 30·8% decline, respectively, in anemia prevalence in women, the largest compared to the other regional states.

## Discussion

This study's findings provide insight into the extent and patterns of child malnutrition and maternal anemia over time in various regions and chartered cities of Ethiopia. Although there

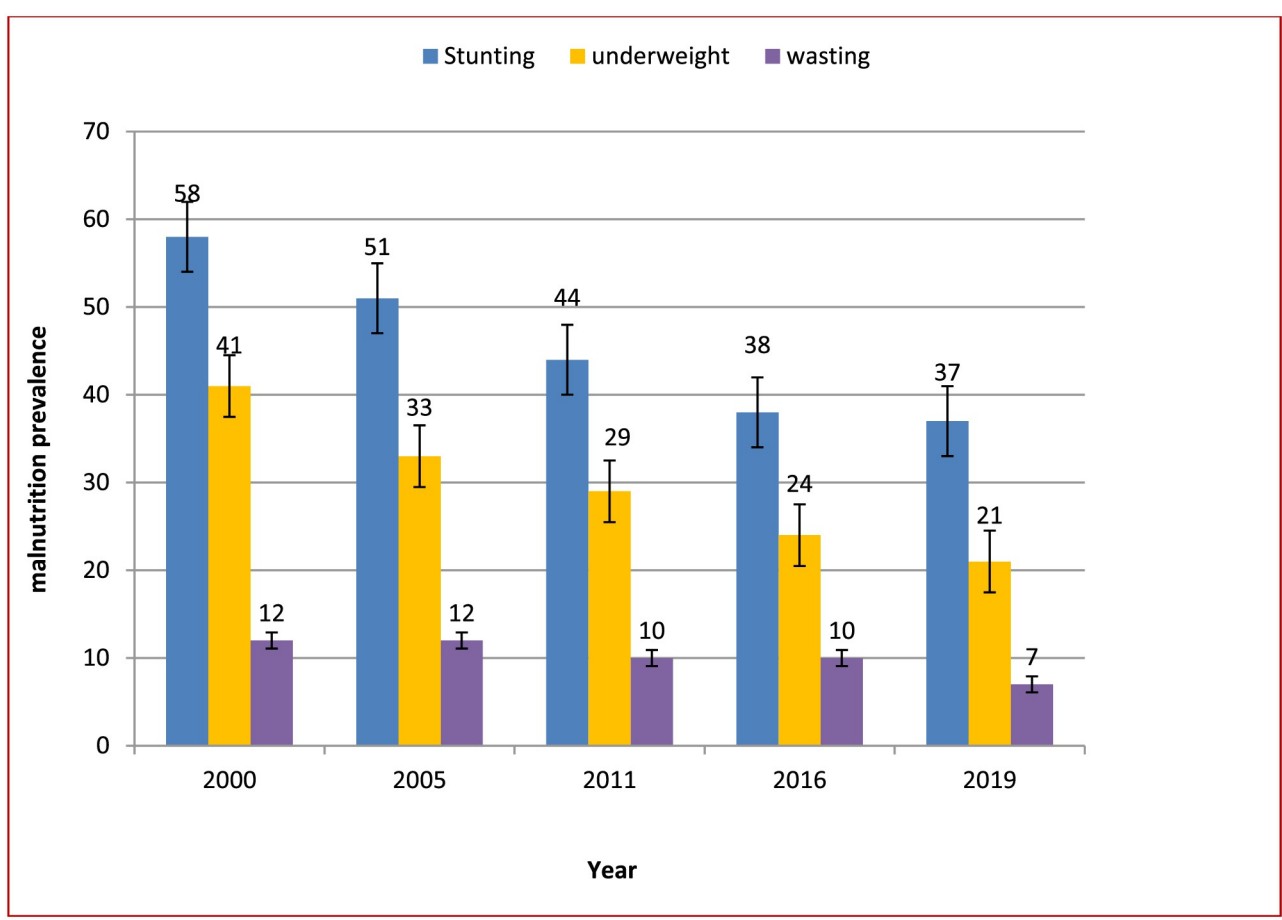

**Fig 4. Trends in the prevalence of malnutrition indicators in children under 5 of both sexes in Ethiopia, 2000–2019.** Note: Malnutrition here refers to undernutrition.

**Table 3. Prevalence of anaemia in children under five years and mothers of reproductive age (15–49 years) in Ethiopian regional states and chartered cities, 1990–2019.**

| | Anaemia prevalence in children | | | | Anaemia prevalence in mothers of reproductive age | | | |
|---|---|---|---|---|---|---|---|---|
| | 1990 (95% UI) | 2019 (95% UI) | Change | % of change | 1990 (95% UI) | 2019 (95% UI) | Change | % of change |
| Addis Ababa | 57.7 (50.7%-64.7%) | 52.2 (46.7%-58.0%) | -5.5 | -9.5 | 19.9 (16.0–24.2) | 13.0 (10.6–15.6) | -6.8 | -34.4 |
| Afar | 73.9 (66.7%-81.0%) | 78.8 (73.1%-84.4%) | 4.9 | 6.6 | 38.7 (35.0–42.3) | 36.1 (32.9–39.5) | -2.6 | -6.7 |
| Amhara | 60.9 (54.3%-67.8%) | 52.2 (46.6%-58.2%) | -8.7 | -14.3 | 25.1 (21.5–28.7) | 16.6 (13.9–19.7) | -8.5 | -33.9 |
| Benishangul Gumuz | 61.9 (54.6%-69.2%) | 52.4 (46.7%-58.5%) | -9.5 | -15.3 | 26.4 (22.7–30.4) | 18.3 (15.4–21.3) | -8.1 | -30.8 |
| Diredawa | 73.3 (68.0%-79.1%) | 74.1 (68.9%-79.3%) | 0.8 | 1.1 | 32.4 (28.8–36.3) | 28.5 (25.1–32.0) | -4.0 | -12.2 |
| Gambella | 63.6 (56.0%-71.2%) | 57.1 (51.2%-63.1%) | -6.5 | -10.2 | 30.7 (26.6–35.0) | 23.0 (19.5–26.4) | -7.7 | -25.2 |
| Harari | 66.2 (59.0%-73.4%) | 67.9 (62.3%-74.2%0 | 1.7 | 2.6 | 25.7 (22.2–29.4) | 23.2 (20.2–26.4) | -2.5 | -9.8 |
| Oromia | 67.2 (60.8%-73.8%) | 67.6 (62.3%-72.8%) | 0.4 | 0.6 | 27.3 (23.7–30.8) | 21.8 (18.9–24.8) | -5.5 | -20.0 |
| Somali | 82.1 (75.4%-88.2%) | 84.4 (79.8%-88.8%) | 2.3 | 2.8 | 44.4 (40.8–48.1) | 45.0 (41.8–48.1) | 0.6 | 1.3 |
| SNNP | 56.1 (49.3%-63.4%) | 52.7 (47.0%-58.8%0 | -3.4 | -6.1 | 22.9 (19.6–26.4) | 17.1 (14.5–20.0) | -5.8 | -25.5 |
| Tigray | 65.2 (57.9%-72.8%) | 58.3 (52.3%-64.9%0 | -6.9 | -10.6 | 22.5 (18.7–26.3) | 16.8 (14.0–20.0) | -5.7 | -25.3 |
| Ethiopia | 63.8 (60.7%-67.1%) | 62.0 (59.1%-65.1%) | -1.8 | -2.8 | 26.2 (24.5–28.0) | 20.4 (19.0–21.8) | -5.8 | -22.1 |

has been a decrease in the prevalence of child malnutrition and maternal anemia at both national and regional levels, these issues continue to be primary contributors to illness and death in children under 5 years and women of childbearing age [27]. Noticeable differences were observed among the regional states, with Somali and Benishangul-Gumuz recording the highest rates of Years Lost due to Disability (YLL). Gambella experienced the most significant reduction in YLL rates across the regions, showing a decrease of 98.2% in stunting, 95.9% in wasting, and 97.9% in underweight from 1990 to 2019. The prevalence of anemia in children under 5 in Ethiopia stood at 62.0%, with no substantial change from 1990 to 2019. The persistent high levels of anemia in children and women, with only a slight reduction since 1990, represent a significant public health concern in Ethiopia.

Since 1990, the Socio-demographic Index (SDI) has consistently risen across various regions, though not uniformly in all areas. City administrations like Addis Ababa and Dire Dawa exhibited higher SDIs compared to other locations. In contrast, pastoralist regions such as Somali and Afar recorded the lowest SDI figures. This disparity could be due to frequent occurrences of drought, floods, and malaria outbreaks in these areas. Additionally, epidemics like cholera and measles, coupled with limited access to healthcare, have contributed to the lower SDI in these regions compared to others. The GBD study done on progress in health among regions in Ethiopia showed the same results [28].

In Ethiopia, the three malnutrition indicators namely stunting, underweight, and wasting has shown a marked decrease over the years, which may be due to different policy interventions, including NNP programs, health sector transformation plans, Seqota declaration, and an increase in the country's socioeconomic status [32]. The major impact of Seqota declaration was a 7.9% absolute reduction or 15.5% relative reduction in stunting in Amhara during the Innovation Phase. The Innovation Phase interventions resulted in a 6.7% absolute reduction or an 18.5% relative reduction in stunting in Tigray and prevented over 1000 deaths in Tigray and Amhara [33]. The overall primary health care expansion and health extension programs were also major interventions that positively impacted the reduction of children and maternal undernutrition in Ethiopia [34]. However, the rate remained high in 2019. Although the improvement in malnutrition prevalence has been marked, it has been much lower than what is needed for Ethiopia to achieve its own NNP 2020 target and the WHO and UNICEF 2030 targets [9]. In addition, the conflict in the years 2021 and 2022 in the northern part of Ethiopia may have had an impact on the high prevalence of child and maternal nutritional status.

The occurrence of stunting in Ethiopia remains high: 37·0% in 2019. This finding was similar to the study finding of the World Bank and a WHO study in 2019, which showed a prevalence of stunting of 36·8% and prevalence of wasting of 7·2% [35]. Stunting prevalence also had extreme variations among the regions. Amhara and Tigray had higher prevalence, with 42·0% and 40·0% respectively in 2016 [21]. This reflects the overwhelming result of malnutrition in early childhood. The effects of childhood stunting are permanent physical and cognitive impairments [36, 37]. This subnational analysis reveals that the capital, Addis Ababa, is the only region on track to achieve a 40·0% reduction in stunting by 2025 [38, 39]. Stunting prevalence showed only a 1·9% annual reduction in Ethiopia from 2000 to 2019, but this decrease was less than the 11·0% annualized reduction needed for the NNP 2020 target and the 14·8% annualized reduction needed for WHO and UNICEF 2025 targets [40, 41]. Wasting prevalence showed a 2·2% annual reduction in Ethiopia from 2000 to 2019, which was well below the 4·8% annualized reduction needed for the NNP 2020 target and the 4·8% annualized reduction needed for WHO and UNICEF 2025 targets [42]. Gambella and Somali are not making any progress in reducing the wasting level in Ethiopia. When compared to other sub-Saharan African countries, Ethiopia showed a modest decline from 1990 to 2019, and urgent intervention is still needed in the country.

Ethiopia has made numerous efforts to tackle child malnutrition over several years, employing different policy measures including the Seqota declaration initiated in July 2015, nutrition programs in rural communities, and the National Nutrition Program (NNP-II) as a strategic guide. Despite these efforts, the rates of stunting, wasting, and underweight children continue to be elevated. Because a decrease in stunting can be accompanied by a transient increase or standstill in wasting, attaining a simultaneous reduction in stunting and wasting can be challenging [43].

The YLL rate attributable to child malnutrition in children reduced by 77·1% between 1990 and 2019 in Ethiopia, a significant reduction. Regional-level differences exist in the YLL rate in all the malnutrition indicators. Stunting, wasting, underweight, and low birthweight showed a substantial decline in YLL rates between 1990 and 2019. YLL rates attributable to child malnutrition decreased between 1990 and 2019 in all regional states of Ethiopia but were low compared to the prevalence of stunting, wasting, and underweight. Somali and Oromia had lower reductions over the past 29 years compared to others regional states. A study conducted in 2019 in Ethiopia on the prevalence of low birthweight strongly supports this finding and noted a prevalence of 14·0%, the highest prevalence in the world [44].

Given that low birthweight is a primary factor in child malnutrition in Ethiopia, prioritizing its gradual reduction is crucial. Nepal, India, and Ethiopia are believed to have the highest rates of low birthweight compared to any other region globally [45]. A major issue in monitoring low birthweight is the poor quality of birthweight data in numerous low- and middle-income countries, Ethiopia included [46]. Low birthweight not only adversely affects the health of children but also increases the likelihood of chronic conditions in later life [47]. The greater percentage of underweight women of childbearing age in Ethiopia, relative to other sub-Saharan African countries, has led to a more prevalent issue of low birthweight in the nation [48].

The prevalence of anemia has been extremely elevated in Ethiopia at 62·0% in children under 5 years of age. The national study conducted by WHO showed 50·4% in 2016 and EDHS 2016 findings 57·0% [49]. An excessive alteration in trends was seen in the prevalence of anemia in the regional states of Ethiopia between 1990 and 2019. Afar, Dire Dawa, and Somali showed an increase of anemia, whereas other states showed a acceptable decline of anemia prevalence in children under 5 from 1990 to 2019. These differences might be accredited to differences in their families' health-care-seeking behaviors and sociodemographic, economic, and nutritional status across the country [50]. Somali had the highest prevalence of anemia in both children and women (84·4% and 45·0%, respectively, in 2019), which requires special intervention. A secondary analysis in Ethiopia between 2011 and 2016 also showed similar findings to this study [51]. The prevalence of anemia in women of reproductive age (20·4%) showed a reduction in all regions of Ethiopia from 1990 to 2019; however, regional-level variation was seen. This finding was higher compared with the national EDHS 2016 finding of anemia prevalence among women of the reproductive age [52].

The major reasons for high occurrence of anemia were poor-quality diets and poor complementary feeding after the age of 6 months in rural parts of Ethiopia [53]. Due to bleeding, anemia in pregnant women might be impacted by parity. Pregnant women who are at dangerous parities and ages for maternal anemia may experience bleeding, which can lead to anemia. Several studies have found a significant association of age and parity with the incidence of anemia in pregnant women [54]. Anemia in women and children is also exacerbated by malaria infection. When malaria parasites enter the bloodstream following a mosquito bite, they infect red blood cells, leading to a reduction in their number. In severe cases, this infection can result in profound anemia [34]. High malaria and intestinal parasites could also contribute to the rise in anemia prevalence in Ethiopia. Variables related to reproductive health and mother's health status (pregnancy status, number of children ever born from the woman, breastfeeding status,

body mass index, use of family planning method, menstrual disorders) may have a great contribution to anemia [55]. Disease and living conditions (including schistosomiasis, hookworm infestations, main source of drinking water for the household, having a mosquito bed net for sleeping) also contributed to anemia in Ethiopia [56].

Anemia increases the likelihood of negative birth outcomes and death during and post-childbirth, and it results in suboptimal cognitive and physical growth and higher mortality rates in children [57]. Ethiopia has launched various programs to tackle the widespread issue of anemia throughout the lifespan, incorporating age-targeted strategies like iron and folic acid supplementation and deworming treatments [58]. As emphasized in Ethiopia's recent NNP-II, a series of initiatives to improve the health of adolescents and young women would be more beneficial than a single macronutrient or micronutrient deficiency solution.

## Implications for research and policy

In Ethiopia, significant improvements in malnutrition indicators will necessitate an integrated nutrition policy that successfully targets the broader drivers of undernutrition throughout the life cycle. This initiative includes ensuring access to clean drinking water, reducing open defecation rates, enhancing women's empowerment in all aspects, boosting agricultural productivity and food security, and advocating for nutrition-focused agriculture. Moreover, a multisectoral approach necessitates collaboration among various ministries and sectors, political commitment, robust governance, and strategic investments [58]. Our findings show that NNP's malnutrition indicator targets for 2020 are aspirational, and that the rate of progress required to reach to these goals is substantially higher than the rate found in this study, making them impossible to achieve in a rapid period of time. Child underweight prevalence showed only a 2·6% annual reduction in Ethiopia between 2000 and 2019. However, this decline fell short of the 8% yearly reduction required to meet the NNP 2020 target [13]. This gradual rate of progress must be boosted so that future malnutrition prevalence figures are better than our estimates based on current trends [59]. The conflict in the northern part of Ethiopia may cause a looming famine and would have a major impact on the efforts to reduce the prevalence of malnutrition. Using the trends displayed for each regional state in these findings, the NNP could set bold but possibly achievable 2030 targets for Ethiopia, just as WHO and UN agencies did after the realization that the SDG target of reducing the level of malnutrition by 2030 was not achievable. Typically, low- and middle-income countries might gain from establishing national and regional objectives for diminishing malnutrition, grounded in a thorough examination of existing trends [59].

## Strengths and limitations

Specific constraints related to the results of this study encompass the somewhat incomplete data on low birthweight in Ethiopia. Frequently, birthweight is either inaccurately recorded or not well remembered by parents, leading to erroneous documentation in numerous cases in Ethiopia. Consequently, household surveys yield relatively limited dependable data on this metric. There is a need for enhancement in data collection methods to procure more accurate estimations of low birthweight [45]. We employed periods to depict the trend changes in malnutrition indicators for their simplicity in comprehension. The main emphasis of this paper was on the undernutrition of women and children in Ethiopia, without evaluating the impact of overnutrition.

Key strength of the conclusions in this report are attributed to utilizing all available data sources in Ethiopia, minimizing the risk of inaccurate estimates often seen in individual surveys with inconsistent data quality [60]. The estimations of malnutrition burden and the

trends for every regional of the country were produced using the standardized GBD methods. The detailed contributions from specialists in Ethiopia regarding the analysis and interpretation of the data are additional strengths of the findings reported in this study.

## Conclusions

The trend in the prevalence of malnutrition was relatively favorable and showed progress. Despite the efforts made over the last few decades, the prevalence of malnutrition and associated morbidity and mortality remain high in Ethiopia. The YLL rate of child malnutrition in all indicators was also very high, though with regional variations. In Ethiopia, there was no notable change in stunting, wasting, and underweight prevalence in the recent two decades. The national nutrition program needs to make more effort to strengthen accessibility and utilization of nutrient-rich foods in collaboration with other sectors, including the agricultural sector. The results in this report give a indication for checking the advancement of malnutrition indicators in the coming years in each region and chartered city. This report's detailed regional assessment of malnutrition indicators, projections, and linkages with policy targets could be beneficial in other similar nations to inform decision-making to eliminate regional variations in nutritional status.

## Acknowledgments

We thank all individuals who have contributed to the GBD 2019 study in numerous dimensions.

## Author Contributions

**Conceptualization:** Mesfin Agachew Woldekidan, Asrat Arja, Mohsen Naghavi, Awoke Misganaw.

**Data curation:** Mesfin Agachew Woldekidan, Asrat Arja.

**Formal analysis:** Mesfin Agachew Woldekidan, Simon Hay.

**Funding acquisition:** Mesfin Agachew Woldekidan.

**Investigation:** Mesfin Agachew Woldekidan.

**Methodology:** Mesfin Agachew Woldekidan, Asrat Arja, Alemnesh Hailemariam, Mohsen Naghavi, Simon Hay, Awoke Misganaw.

**Project administration:** Mesfin Agachew Woldekidan.

**Resources:** Mesfin Agachew Woldekidan.

**Software:** Mesfin Agachew Woldekidan.

**Supervision:** Mesfin Agachew Woldekidan, Nicholas J. Kassebaum.

**Validation:** Mesfin Agachew Woldekidan.

**Visualization:** Mesfin Agachew Woldekidan, Asrat Arja, Simon Hay, Awoke Misganaw.

**Writing – original draft:** Mesfin Agachew Woldekidan, Awoke Misganaw.

**Writing – review & editing:** Mesfin Agachew Woldekidan, Asrat Arja, Getaye Worku, Ally Walker, Nicholas J. Kassebaum, Alemnesh Hailemariam, Mohsen Naghavi, Simon Hay, Awoke Misganaw.

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
