## [Decision Letter · Decision Letter 0]

12 Sep 2023

PGPH-D-23-01229

The burden and trends of child and maternal malnutrition across the regions in Ethiopia, 1990-2019: The Global Burden of Disease Study 2019

Dear Dr. Woldekidan,

Thank you for submitting your manuscript to PLOS Global Public Health. After careful consideration, we feel that it has merit but does not fully meet PLOS Global Public Health’s publication criteria as it currently stands. Therefore, we invite you to submit a revised version of the manuscript that addresses the points raised during the review process.

We look forward to receiving your revised manuscript.

Kind regards,

Sirshendu Chaudhuri, MD, DPH

Academic Editor

Journal Requirements:

1. We noticed you have some minor occurrence of overlapping text with the following previous publication(s), which needs to be addressed:

- https://www.researchgate.net/publication/335888155_The_burden_of_child_and_maternal_malnutrition_and_trends_in_its_indicators_in_the_states_of_India_the_Global_Burden_of_Disease_Study_1990-2017

- https://www.thelancet.com/journals/lanchi/article/PIIS2352-4642(19)30273-1/fulltext

- https://bmcpublichealth.biomedcentral.com/articles/10.1186/s12889-019-8019-z

- https://www.jemds.com/data_pdf/ajitha-prav-Ori-ECC-.pdf

- https://sussex.figshare.com/articles/journal_contribution/Progress_in_health_among_regions_of_Ethiopia_1990_2019_a_subnational_country_analysis_for_the_Global_Burden_of_Disease_Study_2019/23488124

In your revision ensure you cite all your sources (including your own works), and quote or rephrase any duplicated text outside the methods section. Further consideration is dependent on these concerns being addressed.

Additional Editor Comments (if provided):

Reviewers' comments:

Reviewer's Responses to Questions

**Comments to the Author**

1. Does this manuscript meet PLOS Global Public Health’s publication criteria? Is the manuscript technically sound, and do the data support the conclusions? The manuscript must describe methodologically and ethically rigorous research with conclusions that are appropriately drawn based on the data presented.

Reviewer #1: Yes

Reviewer #2: Yes

2. Has the statistical analysis been performed appropriately and rigorously?

Reviewer #1: Yes

Reviewer #2: No

3. Have the authors made all data underlying the findings in their manuscript fully available (please refer to the Data Availability Statement at the start of the manuscript PDF file)?

Reviewer #1: Yes

Reviewer #2: No

4. Is the manuscript presented in an intelligible fashion and written in standard English?

Reviewer #1: Yes

Reviewer #2: No

5. Review Comments to the Author

Reviewer #1: Child malnutrition and maternal anemia are major health concerns in Ethiopia. This study analyzes data from 1990 to 2019, revealing high prevalence of stunting, underweight, wasting, and anemia in children. Child malnutrition contributed to a significant portion of under-five deaths. Despite interventions, malnutrition remains above global targets, requiring sustained efforts, including nutrition initiatives and improved food access. This study is well discussed to provide the proper interpretation and usefulness of this paper. Here, the paper has well written and includes better exciting research work. So, I am completely recommended the article for publication.

Reviewer #2: Thank you so much for giving me the opportunity to review this study. Thanks the authors for their hard work with this manuscript.

In the title, the authors mentioned- “maternal malnutrition”. However, I see only anaemia was dealt with. Suggest restricting the title to anaemia instead of ‘malnutrition’ which is a broader term.

The introduction is quite large. More focus can be paid to methods (For even more clarity).

About the data source: The authors mentioned that- “The details on data sources and methods in Ethiopian subnational GBD analysis were reported on the health progress study in Ethiopia(30).” However, I feel it is quite relevant that you provide a brief summary of the data source, and how that data was collected. How the authors retrieved it and cleaned it. This is important as it is directly related to the outcome of interest. For example, I understand that stunting definition is <-2SD of height for age. But how the height and weight was measured? How did you calculate the standard deviation (The software). Did you exclude any outlier? Or, a readymade calculation was available and you used it directly. That kind of granularity should be available in the methods.

It will be nice if you can add a section on human subject protection even if it is a secondary data source.

The authors used the term malnutrition synonymously with undernutrition, which is actually incorrect. Is there a reason why they are not using ‘undernutrition’ straightaway?

Table 3: ‘Mothers’ or women of reproductive age?

There are some typo errors in the table. Please revise

The authors themselves expressed their concern about the source data in the limitation section. This is critical for the entire paper and I expressed my concern earlier. The authors must ensure adequate information about the data management and what kind of bias they expect, they should state clearly. The authors suggested in the same section what can be done in future. As a reviewer, I am concerned about the reliability of the present finding. Please focus there. Again, in the same place, they mentioned about what was not done. Please stick to what you have done.

Finally, the paper needs some brevity, and some improvement in the language would be wonderful.

6. PLOS authors have the option to publish the peer review history of their article (what does this mean?). If published, this will include your full peer review and any attached files.

**Do you want your identity to be public for this peer review?** For information about this choice, including consent withdrawal, please see our Privacy Policy.

Reviewer #1: No

Reviewer #2: **Yes: **Sirshendu Chaudhuri

---

## [Decision Letter · Decision Letter 1]

30 May 2024

The burden and trends of child and maternal malnutrition across the regions in Ethiopia, 1990-2019: The Global Burden of Disease Study 2019

PGPH-D-23-01229R1

Dear Woldekidan, 

We are pleased to inform you that your manuscript 'The burden and trends of child and maternal malnutrition across the regions in Ethiopia, 1990-2019: The Global Burden of Disease Study 2019' has been provisionally accepted for publication in PLOS Global Public Health.

Best regards,

Santosh Kumar

Academic Editor

**Santosh Kumar, Ph.D.**

Associate Professor of Development and Global Health Economics

Keough School of Global Affairs

Concurrent faculty, Department of Economics

University of Notre Dame

Notre Dame, IN 46556

Reviewer Comments (if any, and for reference):

Reviewer's Responses to Questions

**Comments to the Author**

1. If the authors have adequately addressed your comments raised in a previous round of review and you feel that this manuscript is now acceptable for publication, you may indicate that here to bypass the “Comments to the Author” section, enter your conflict of interest statement in the “Confidential to Editor” section, and submit your "Accept" recommendation.

Reviewer #1: All comments have been addressed

2. Does this manuscript meet PLOS Global Public Health’s publication criteria? Is the manuscript technically sound, and do the data support the conclusions? The manuscript must describe methodologically and ethically rigorous research with conclusions that are appropriately drawn based on the data presented.

Reviewer #1: Yes

3. Has the statistical analysis been performed appropriately and rigorously?

Reviewer #1: Yes

4. Have the authors made all data underlying the findings in their manuscript fully available (please refer to the Data Availability Statement at the start of the manuscript PDF file)?

Reviewer #1: Yes

5. Is the manuscript presented in an intelligible fashion and written in standard English?

Reviewer #1: Yes

6. Review Comments to the Author

Reviewer #1: Thanks to the author to revise the manuscript as per reviewers comments.

7. PLOS authors have the option to publish the peer review history of their article (what does this mean?). If published, this will include your full peer review and any attached files.

**Do you want your identity to be public for this peer review?** For information about this choice, including consent withdrawal, please see our Privacy Policy.

Reviewer #1: No
